# Human- common hippo (*Hippopotamus amphibius*)-conflict in the Dhidhessa Wildlife Sanctuary and its surrounding, Southwestern Ethiopia

**Girma Gizachew Tefera**[1]*, **Tadesse Habtamu Tessema**[2], **Tibebu Alemu Bekere**[1], **Tariku Mekonnen Gutema**[1]

1 Department of Natural Resource Management, Colleague of Agriculture and Veterinary Medicine, Jimma University, Jimma, Ethiopia, 2 Department of Biology, College of Computational Sciences, Jimma University, Jimma, Ethiopia

* kennaa20047@gmail.com

**Data Availability Statement:** All relevant data is within the paper and it's supporting information files.

## Abstract

The common hippopotamus (Hippopotamus amphibius) is one of the most endangered mega herbivore in Africa. Although the human-hippo conflict (HHC) is currently escalating due to habitat loss, little is known about the scope of the conflict and potential mitigation measures. From 2021 to 2022, the extent of HHC, the perception of local inhabitants towards the animal, and its impacts on the conservation of the hippo were assessed within and in the surrounding areas of Dhidhessa Wildlife Sanctuary, southwestern Ethiopia. A total of 227 households were used as a data sources, participating in the questionnaire survey, focus group discussion and key informant interviews. Direct field observations were also used as key data source. Crop raiding and damage (63%, n = 143), was the most severely reported source of conflict, followed by direct human attack (22.9%, n = 52). Livestock predation and increased hippo mortality were also common types of HHC in the area. According to the study, the majority of respondents (74.4%, n = 169) agreed that the trend of conflict was increasing, while 16.7% (n = 38) argued that there was no change. Food scarcities in the buffer zone, as well as agricultural expansion in the area, have been identified as major drivers of HHC. The majority of respondents had a negative attitude toward hippo conservation; however, there were differences based on gender, age, and educational level. The major traditional hippo conflict mitigation strategies in the area include guarding, fencing, and trenching. Field observation revealed that a large extent of the former grazing and ranging areas of hippo in the area are heavily encroached and taken over by large private and public agricultural investments. Vacating and recovering at least the former buffer areas dedicated to the wildlife in the area and modern, non-lethal mitigation strategies are recommended for better conservation and safeguarding of the currently good-sized hippo population of the sanctuary.

**Funding:** The author(s) received no specific funding for this work.

**Competing interests:** The authors declare no conflict of interest

## Introduction

Human-wildlife conflicts (HWC) are a common global phenomenon and more sever in areas where humans and wild animal's requirements overlap [1, 2]. The conflicts cause an adverse effect both on humans (economic, cultural, and well-being) and on wildlife survival and conservation [3, 4]. Crops raiding, livestock depredation, destruction of fishery tools, and human attacks are common causes of the conflicts [5, 6]. Fast human population growth, land use change, species habitat loss, and global climate changes are reported aggravating factors for the current high incidence of human-wildlife conflicts in Africa [7–10].

The common hippopotamus (*Hippopotamus amphibius*) is the third largest mega herbivore, currently is becoming a key candidate for such negative human wildlife interaction. Formerly, this species was roaming in wetlands of most sub-Saharan Africa; however, latest estimates [11, 12] predicted between 7 and 20% population decline, range reduction and habitat fragmentation during the last century and currently, the species is listed as "vulnerable" on the IUCN Red List [13]. For the threats from habitat loss and degradation, poaching and trade in hippo parts (teeth, skulls, ivory, skin and meat) and largely, for the fast expanding human-hippopotamus conflict (HHC) [5, 14] the species is exposed to a high risk of extinction.

Like most other hippo range states in Africa, rivers, lakes and wetlands in most parts of Ethiopia provide ideal habitat, and the species are fairly common in these ecosystems. These days, however, for the stated reasons, many of the wetlands that used to serve as feeding and breeding areas of this species have been drained for seasonal subsistence farming, and the former wilderness areas following river basins are converted to mechanized commercial farms and heavily encroached. Among the several river basins in Ethiopia, the Dhidhessa river basin, covering most of the southwestern part of the country, contains the largest hippo concentration in the country. As a result, in 1970, the Ethiopian government designated the area as Dhidhesa Wildlife Sanctuary, particularly to conserve the area for the species. Like most challenges of the wildlife conservation areas in the country, the ideal hippo ecosystem in this sanctuary were heavily encroached by settlement of drought displaced communities from the northern and eastern part of the country. In addition, the adjacent areas around the sanctuary were converted into mechanized sugarcane farm growing row material for the Arjo-Dhidhesa Sugar factory.

For over the past two decades, reports on HWC studies in Ethiopia focus only on few very common pest species including olive baboon (*Papio Anubis*, 21 studies), porcupine (*Hystrix cristata*, 17 studies), spotted hyena (*Crocuta crocuta*, 15 studies), leopard (*Panthera pardus*, 12 studies), and common warthog (*Phacochoerus africanus*, 10 studies) [15] and also the majority of them from National Parks [5, 16].

Recently, Dhidhessa Wildlife Sanctuary has not provided the intended objectives for which it was established. Subsequently, large mammals that occurred in the area migrated into the nearby forest patches. But the hippo is one of many large mammals that have been living in the Dhidhessa River valley. However, in the study area, no comprehensive research has been done on the causes, extent, and consequences of HHC in the southwestern regions of Ethiopia.

In this regards, a better understanding of the causes of HHC in this unique area is essential for the development and implementation of effective mitigation strategies. Therefore, the purpose of this study was to examine the causes, extent, and impacts of HHC and to explore the attitudes and conservation interests of local inhabitants towards hippo in order to devise plausible conservation management and mitigation strategies in this very important stronghold of hippo- dominated wildlife sanctuary in Ethiopia.

## Materials and methods

### Ethical statement

The research proposal (S1 Appendix) was examined and authorised by the Jimma University (JU) College of Agriculture, Veterinary, and Medicine Review Board. The request was reapproved and accepted by the vice president's office review board for research and community service. JU also accepted the consent -collecting process for the questionnaire (S1 Appendix). Following clearance of the human ethics research, we received a letter of authorization along with a request for cooperation from all Kebeles to carry out this research project in the vicinity of Dhidhessa Wildlife Sanctuary.

**Ethical clearance.**   The University of Jimma provided ethical clearance for this work, with the reference number R/GS/752/2021.

### Description of the study area

This study was conducted in Dhidhessa Wildlife Sanctuary (DWS) and its surrounding areas. The Sanctuary is located at 395 km from the capital, Addis Ababa (through Nekemte route), and situated between 8°30' and 8°40' N latitude and between 36°22' and 36°43' E longitude and covers an area of 1300 km$^2$ shared among three administrative zones (East Wollega, Buno Bedele, and Jimma), all within Oromia National Regional State (Fig 1). The altitudes within the sanctuary range between 1350 and 1050 masl. The Sanctuary experiences two distinct season, dry (November through February) and wet (June through September) and receives a unimodal rainfall varying from 648 and 2001.8 mm. Temperature of the lowland areas in the sanctuary is relatively wormer with a mean annual minimum and maximum temperatures of 12 and 35°C, respectively, [17]. Topography of the area comprised moderately undulating, hilly and mountainous areas with flat savanna and marshy plains towards the river bank. Most habitats in the area include wooded grassland along the hill sides, marsh mangrove, riverine vegetation along the Dhidhesa River and its perennial tributaries and the extensive flat savanna grassland and seasonal marshy areas towards the river banks. The area used to harbor a diversity of wildlife species (especially antelopes, mega herbivores and large carnivores) for which it was designated as Dhidhesa Wildlife Sanctuary, in 1970, mainly for the conservation of the exceptionally dominant species, *Hippopotamus amphibious*. During the consecutive decades, for multiple reasons (including the political instability and recurrent droughts), the majority areas of the Sanctuary, before it was legally gazetted, was heavily encroached and used for settlement for drought displaced people from the North and Eastern part of the country, while the remaining adjacent plains were tailored for mechanized sugarcane farm to produce row material for the state owned Arjo-Dhidhessa Sugar Factory (ADSF). As a result, the Dhidhessa Wildlife Sanctuary failed to provide the objectives for which it was established.

Even so, the area is not officially protected at the moment because it is heavily utilized by the private sector and local communities for mechanized monoculture cane farms, subsistence farming, husbandry, and other economic activities that put them in direct conflict with the hippo and other wildlife species in the area. Nevertheless, it remains an important area for biodiversity conservation in the southwestern parts of Ethiopia. In addition to hippos, DWS and the adjacent surroundings are home to a variety of animal species, including primates (*Papio anubis* and *Cercopithecus mitis*), Buffalo (*Syncerus caffer)*, Lion (*Panthera Leo*) Guerza (*Colobus guerza*),Warthog (Phacochoerus africanus), Bushpig (*Potamochaerus larvatus*) and Spotted Hyena (*Crocuta crocuta*) [20]

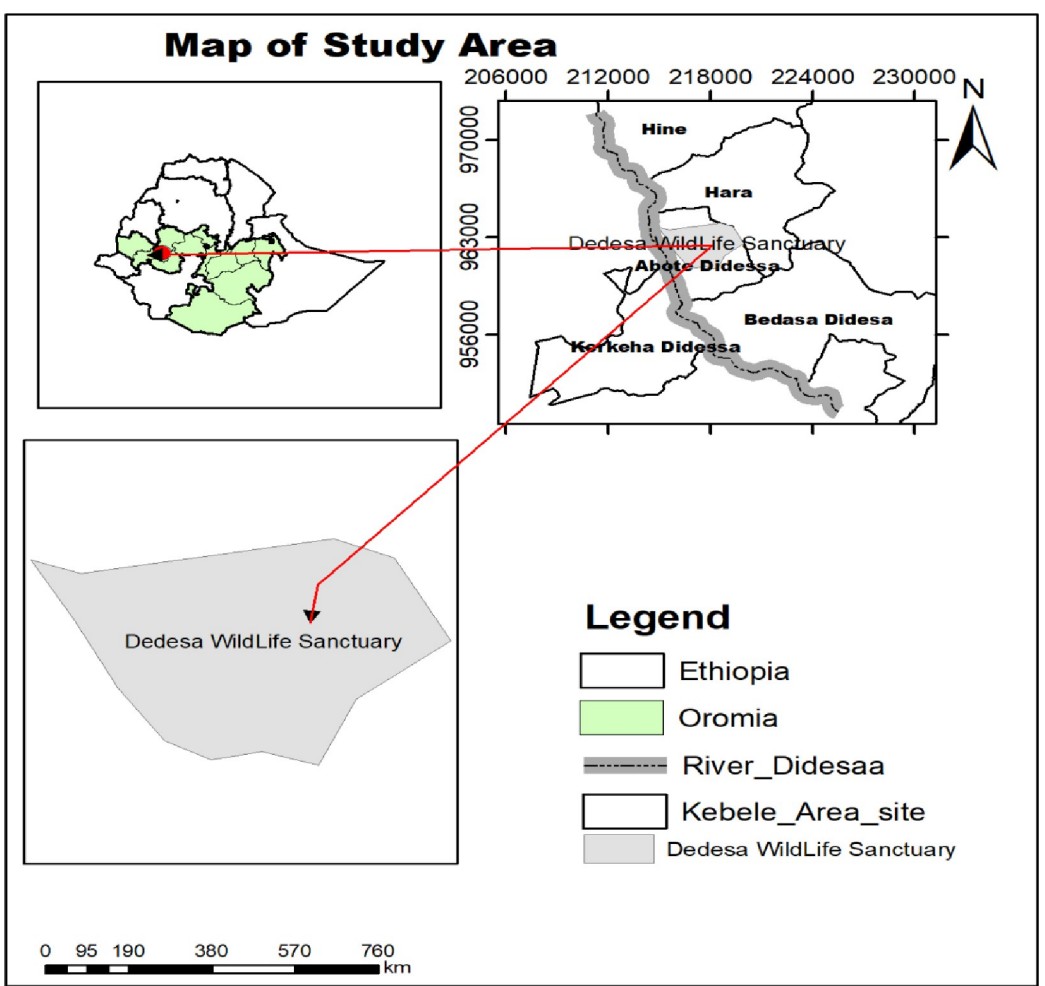

**Fig 1. Map of study area.** (Our team used shape files provided by the Ethiopian Mapping Agency to produce the map. The agency's official website offers free access to these shape files (http://www.ema.gov.et).

## Site selection and sampling design

A preliminary survey was conducted in mid-October 2021 to identify the boundaries, decide the number of villages, and gain a general understanding of the overall situation of the sanctuary. In order to assess the resource sharing conflict between humans and hippo, semi-structured questionnaires (open-ended) were administered to the farmers. The questionnaires were pre-tested among some groups of respondents, which did not include the main sample group. Five villages and a sugarcane plantation were purposefully selected based on the distance from the sanctuary, problems related to crop damage, and conservation challenges in the area. These villages were Bedasa Dhidhessa, Abote Dhidhessa, Hine, Hara, Kerkeha Dhidhessa, and Arjo Dhidhessa Sugar Factory. On the other hand, factors such as the size and proximity of villages to the sanctuary, the river bank, and farm extents were considered for the selection. In terms of their distribution and settlers size, every village was unique. Abote Dhidhesa and Kerkeha Dhidhesa villages are located within two kilometres from the sanctuary; Hine and Hara villages are located between three and five kilometres away from the sanctuary. The Bedasa Dhidhesa and the Sugar Farm Factory are immediately adjacent to the sanctuary. Two hundred twenty-seven (227) households were randomly selected from these five villages and the Arjo Dhidhessa Sugar

Factory. The total population of households' sample frame was established by collecting a complete landholders' list record from each village's administration office. Based on the [18] population correction factor, 227 household heads (154 male and 73 female)—about 20% of the total number of households (3239 households)—were randomly selected.

## Data collections methods

We collected data from February 2021 and July 2022. Prior to the data collection, we obtained permission from the environmental authorities and local administrative offices to conduct the research. We explained that the research was intended only to gather genuine, relevant information to help support the hippo conservation effort of the DWS, balancing the needs of the people and the hippo.

We employed questionnaire survey, key informant interviews, focus group discussion, direct observation (see S1 Appendix), and secondary data collection to investigate HHC in DWS, following [18]. We also requested the consent of respondents to participate in the household interview, and only voluntary respondents were interviewed. Researchers collect data from participants by using the Kobo Toolbox, tape recorders, and note-taking. KoBoToolbox is a suite of tools for field data collection for use in remote areas or where the internet facility is uncertain. (https://support.kobotoolbox.org/). We collected quantitative data through close-ended and open-ended questionnaire interviews and qualitative data through focus group discussions and key informant interviews. We designed the questionnaire to assess the cause, extent, and impacts of HHC, the general socioeconomic status of the community, the attitudes of people towards hippopotamus conservation, and protection measures adopted to protect crop damage.

To complement the household survey, we collected basic qualitative (descriptive) information through a series of focus group discussions. Three focus group discussions were conducted to explore their experiences with human-hippo conflicts and the associated impacts on their livelihood. The group sizes in each discussion group varied from 5 to 10 people. In addition, we selected 10 key informants based on their field of work, experience, and age. Key informants were zone and sugar factory environmental experts, wildlife and forest experts, Woreda agricultural office leaders, Kebele administrators, Woreda natural resource protection authorities, and elders in consultation with Woreda and Kebele administrators. This enabled us to obtain qualitative data through in-depth interviews and discussions with each respondent using an unstructured questionnaire. We conducted all interviews and focus group discussions in a local language, Oromifa.

Furthermore, we completed direct observations by trekking through the study sites during the day when humans, livestock, and hippo are active. Also, to further confirm and strengthen the authenticity of the collected data, a two-year (2020–2021) on-farm direct observation was conducted to record crop raids or trembles during the crop seasons. All incidences of fresh hippo attacks, injuries to humans and domestic animals, and farmer strategies to lower the conflict and lethal revenge strategies were recorded. In addition, ten years of secondary data (2014–2022) were accessed to estimate the severe casualties among humans in the area.

To observe the extent of crop damage by hippo and to compare the result with the response given by the local people, four sites were chosen at random. Four equivalent, 28,000-square-metre-farmed patches were randomly chosen for each site. According to the method adopted by [16], every crop area was split into four grids, each with a 7,000 m2 area. Crop species consumed, area of damaged portion, part of plant eaten, and type of crop cultivated were all recorded for each grid [22]. Every three days, farming grounds were inspected, and, among other things, crop damage was noted. The damaged portions were immediately calculated in

square m2, and the grids comprised regions with sown crops. The farm's distance from the river tainted with hippos, its area, and farms on or near animal movement routes; the crop-growing season; its proximity to other farms or settlements; and the farm's available farm guarding techniques from the crop field to the sanctuary were also recorded [22].

## Data analysis

To analyze the gathered data, descriptive statistics and the Pearson chi-square test were employed. The significant differences between the villages in terms of crop damage prevention strategies, crop damage trends, and the locals' attitudes towards hippo, and the kinds of conflicts they had with hippo were assessed using a Pearson chi-square test. Significant data was obtained using a chi-square test with a 2-tailed $P<0.05$. Cramer's V was used to determine the degree of statistical significance ($P< 0.05$) for the correlations. Information gathered from focus group discussions was analyzed to create a summary before being presented narratively. All analyses were conducted using the Statistical Package for Social Sciences (SPSS version 25, Chicago, USA).

## Results

### The demography and socio-economic activities

Out of 227 respondents to the household survey, 67.8% (n = 154) were male and 32.2% (n = 73) were female. Regarding educational level, most (70%, n = 159) were literate, and the rest (30%, n = 68) were uneducated. In terms of age, 32.6% (n = 74) and 40% (n = 90) of respondents in the study area were in the age groups of 25–34 and 35–50, respectively. Relatively few (13%, n = 30) of respondents were equal to or older than 50 years of age. In general, more than 73% of respondents in the study area were in the age groups of 25–50. Most of the respondents (75.3%, n = 171) were involved in crop farming and livestock rearing, while 24.7% (n = 56) were daily laborers or employees (Table 1). The most commonly grown crops in the area include maize, sorghum, sugar cane, wheat, groundnut, and vegetables, with cattle, sheep, and goats being the most common livestock held by the small holders.

### Human-hippo conflict

Among the respondents, 63% (n = 143) reported that crop damage was the major cause of human-hippo conflict, followed by physical threats to humans (n = 52). While 3.1% (n = 7) of the respondents reported that livestock mortality also contributed to the study area. During this study, no clear hippo-crop preference was observed; however, maize (49.8%, n = 113), sugar cane (24.2%, n = 55), and sorghum (22.5%, n = 51) were the most frequently visited crops in the area (S1 Table in S2 Appendix). Grazed plant remains and trampled crops were major evidences for crop damage (35.2%); hippo hoofmark (30.4%); and dung pile (13.2%) were also used for confirmation (S1 Fig in S2 Appendix). The spatial distributions of conflict incidences show variation among sampled study sites, where areas including Bedessa, Dhidhessa, and Arjo Dhidhessa Sugar Factory had the highest incidence of crop damage, while Hinne and Hara had the least, and the differences are highly significant ($X^2$ = 32.57, df = 20, $p<0.05$) (Fig 2).

Responses of informants regarding the impacts of hippo on livestock are of two type, transmitting a deadly cattle disease called African trypanosomiasis (TA) (locally called Gendi) (32% = 63), and through excluding livestock from important seasonal pasture (12%, n = 42). The results indicated that respondents throughout each surveyed village reared about 2368 cattle.

**Table 1. Demography and socio-economic activities of the respondents in the study area.**

| Variable | Frequency | Percentage (%) | Mean | Standard deviation |
|---|---|---|---|---|
| Sex | | | 1.32 | 0.468 |
| Male | 153 | 67.8 | | |
| Female | 74 | 32.2 | | |
| Age (year) | | | 2.52 | 0.899 |
| 18–24 | 33 | 14.5 | | |
| 25–34 | 74 | 32.6 | | |
| 35–50 | 90 | 40 | | |
| >51 | 30 | 13 | | |
| Education level | | | 2.2 | 1.106 |
| Illiterate | 68 | 30 | | |
| Primary school | 97 | 42.7 | | |
| Secondary school | 10 | 4.4 | | |
| College and above | 52 | 22.9 | | |
| Family size | | | 1.97 | 0.857 |
| 1–3 | 74 | 32.6 | | |
| 4–6 | 97 | 42.7 | | |
| 7–9 | 44 | 19.4 | | |
| >10 | 12 | 5.3 | | |
| Marital status | | | 1.24 | 0.671 |
| Married | 196 | 86.3 | | |
| Single | 13 | 5.7 | | |
| Divorced | 12 | 5.3 | | |
| Widowed | 6 | 2.6 | | |
| Livelihood source | | | 2.46 | 0.903 |
| Mixed agriculture | 163 | 71.8 | | |
| Only farming | 8 | 3.5 | | |
| Wardens/ employee | 56 | 24.7 | | |
| Livestock size | | | 1.16 | 0.423 |
| 1–10 | 170 | 74.9 | | |
| 11–20 | 37 | 16.3 | | |
| 21–30 | 20 | 8.8 | | |

Nearly 27.7% (n = 655) of the cattle from all the villages surveyed died as a result of the fatal infectious disease between 2013 and 2022.

In this study area, major crop damage was recorded during the night (76.2% n = 173) and more during the rainy seasons (June-August). Maize (the most raided crop) was grazed most at seedling (50.7%, n = 115) and at the vegetative growth stage before flowering (47.1%, n = 107) (see S3 Appendix).

Crop susceptibility to hippo attack depends on a number of factors, including farm size (Mann-Whitney U = 1528.5, p = 0.000), the distance of the farm from the Dhidhesa River (Mann-Whitney U = 367.5, p = 0.0.003), and the distance of hippo access points from the farm (Mann-Whitney U = 408, p = 0.014). However, farm size and distance from the River had the greatest impact (Table 2).

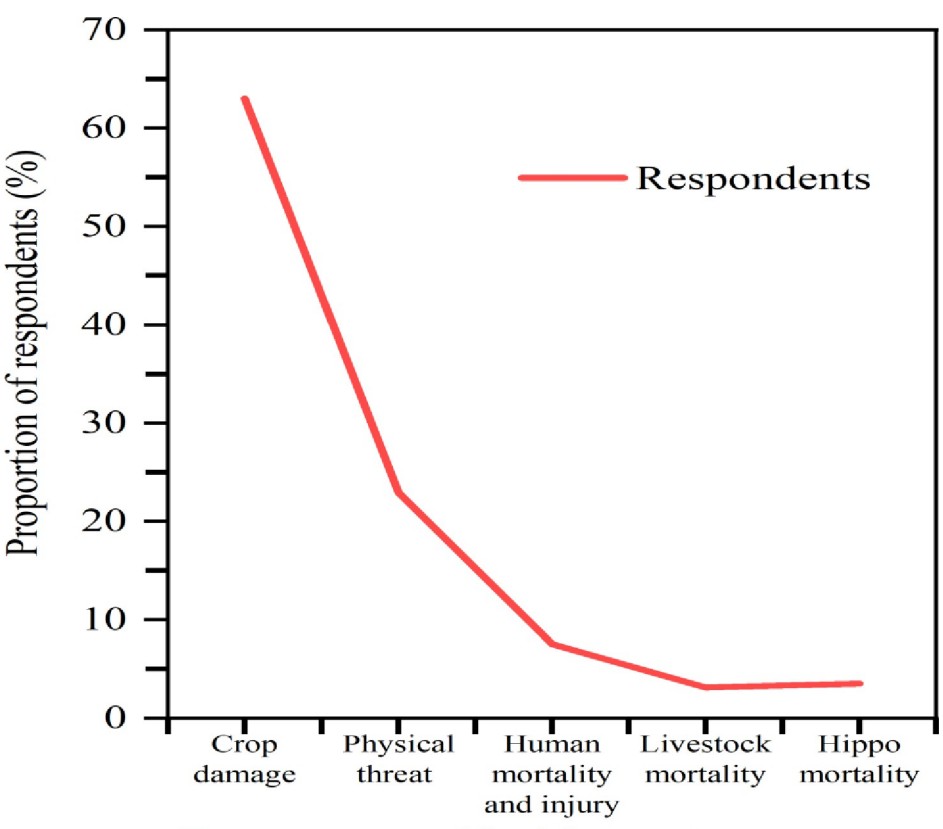

**Fig 2. Percentage of respondents reporting damages caused by hippopotamus.**

### People's perceptions and attitudes towards hippo

Based on the respondents, the occurrence and frequency of crop raiding are dependent on a multitude of conditions, such as the level of human activity on the farm and cultivated crops with respect to the sanctuary location. Out of 227 household respondents, 88% (n = 199) reported that the trend is increasing over time. However, 4% (n = 9) of them said it is decreasing, and the rest 3% (n = 8) reported that there is no idea about the trend of crop raiding. There were significant variations in the pattern of hippo-caused agricultural damage among the villages ($\chi2$ = 33.35, df = 15, P<0.05). Nearby residents of the sanctuary, such as Bedessa

**Table 2. Spatial factors of farms with and without occurrence of hippopotamus crop raiding.**

| Factors | Group | N | Mean | SD | SE | U | Z | P |
|---|---|---|---|---|---|---|---|---|
| Distance of River from farm (km) | With farms | 39 | 1.128 | 0.568 | 0.091 | 367.5 | -2.93 | 0.003** |
| | Without farms | 31 | 0.776 | 0.287 | 0.052 | | | |
| Distance to hippopotamus access points from farm (km) | With farms | 39 | 0.963 | 0.445 | 0.071 | 408 | -2.46 | 0.014 |
| | Without farms | 31 | 0.700 | 0.220 | 0.040 | | | |
| Size of farm (ha) | With farms | 115 | 7.65 | 5.092 | 0.475 | 1528.5 | -4.83 | 0.000** |
| | Without farms | 50 | 3.37 | 1.758 | 0.249 | | | |

* Values with farms mean having cultivated crops, while values without farms mean non-cultivated crops.

Dhidhessa (96%), and Arjo Dhidhessa Sugar Factory (100%), saw the greatest crop loss in comparison to respondents who live two kilometers from the DWS (Table 3).

The majority of the respondents (53.7%, n = = 122) believe that the continual decline of available grazing areas is the main cause for human-hippo conflict. The abundance of crops on farmland (23.3%, n = 53), increased human population (19.8%, n = 45), and lack of effective hippo control system (3.1%, n = 7) were also contributing to the rising conflict (S2 Fig in S2 Appendix). In this study, there were no discernible opinion differences among villagers for the causes of conflict ($\chi2$ = 10.2, df = 15, P>0.05). The Cramer's V statistic was found to be 0.122, indicating that the data can account for 12% of the variations in conflict causes among villages (Fig 3).

Of the total respondents, the majority (70%, n = 159) of them had a negative attitude towards the hippo found in the study area. In contrast, 25% (n = 56) of the respondents had a positive attitude towards hippo. The remaining 5% (n = 12) of the respondents had neither a positive nor negative attitude towards hippo (Table 4). More than half of respondents from villages distant from the sanctuary, such as Hara (51.3%, n = 20) and Hinne (64%, n = 16), had a positive attitude towards hippo, while respondents closest to the sanctuary (Bedhessa Dhidhessa = 87.7%, n = 43, and Abote Dhidhessa = 85.7%, n = 24), had a negative view towards hippo. There was a significant difference in the attitude towards wildlife among respondents in different villages ($\chi2$ = 58.95, df = 10, $P$<0.05). The result shows that men were comparatively more (22.5%, n = 51) supportive of hippo conservation than their female counterparts (5.7%, n = 13), with a significant difference ($\chi2$ = 10.48, df = 2, p = 0.03). In addition, a younger group of the society (21.6%, n = 49) had a more positive attitude towards hippo conservation than older groups (7.5%, n = 17), and the variation was significant ($\chi2$ = 36.12, df = 6, p<0.05). Respondents with a higher level of education (18%, n = 40) had a relatively better understanding and positive attitude than those with a lower level of education or who were illiterate, and this difference was statistically significant ($\chi2$ = 90.2, df = 6, p<0.05).

The Cramer's V test for the relationship between gender and perception towards hippo conservation was 0.21, while those with age and level of education were 0.23 and 0.36, respectively, indicating that about 21%, 23%, and 36% of the variability in hippo perception were explained by gender, age, and level of education, respectively. Both findings indicated that views of gender and age towards hippo were moderately correlated, whereas education level had a medium association with perceptions towards hippo conservation.

## Consequences of conflict both on humans and hippo

During this study, interviews and focal group discussions were held with settlers residing in the area for between two and four decades. The majority (62.1%, n = 141) of respondents informed us that, before two decades, human-hippo interaction was harmless, to the extent that people swam in the same pool infested by hippo along the Dhidhessa River (S2 Fig in S2 Appendix). Vocally distracting hippos not to approach farms was also a common mitigation strategy. Sharing local resources, such as cutting wood for construction and collecting it for firewood, and fetching water were very common in the area, while hippo herds engaged in their daily activities. However, hippo behaviour significantly changed from early 2014 and gradually turned highly aggressive, and the interaction between humans and hippos completely transformed and became what is observed today, where one completely avoids the presence of the other. Most (83.6%) interviewees stated (S3 Fig in S2 Appendix) that the major reasons stated were the aggressive settlement programme by the government, land use change (allocation of a large extent of hippo land to a mechanised state farm), illegal encroachment that restricted and blocked movement corridors and ranging routes of the animals, and

**Table 3. Approximate distance from the sanctuary and trend in crop damage by hippopotamus in the last five years.**

| Villages | N | Distance from the sanctuary (km) | Trends of crop damage compared to the last five year (%) | | | |
|---|---|---|---|---|---|---|
| | | | Increased (%) | Decreased (%) | Stayed the same (%) | Unsure (%) |
| Bedessa Dhidhessa | 49 | 1 | 96 | 0 | 2 | 2 |
| ADSF | 56 | 1 | 100 | 0 | 0 | 0 |
| Abote Dhidhessa | 28 | 1.1–2 | 89 | 7 | 0 | 4 |
| Kereka Dhidhessa | 30 | 2.1–3 | 90 | 0 | 7 | 3 |
| Hara | 39 | 3.1–4 | 74 | 8 | 10 | 8 |
| Hinne | 25 | 4.1–5 | 60 | 16 | 20 | 4 |
| Total | 227 | | 88 | 4 | 5 | 3 |

*ADSF = Arjo Dhidhessa Sugar Factory

repeated incidences of negative encounters that, as a result, drove the animal to be more anxious. Then, hippos are forced to wander into settlements and farm areas and experience aggressive grazing, trembling, and damaging farms.

Of the total respondents, the majority (68.7%, n = 156) interviewees did not appreciate the presence of hippos in their localities and also stated that hippos do not provide any economic or ecological benefits; rather, they are negatively impacting their way of life. As a result of the new negative interaction, the death rate of hippos was continually increasing year after year. Although accessing recorded data for hippo mortality and human fatalities was difficult, informants reported that between 2013 and 2022, there were at least 29 hippo deaths (Fig 4) and 36

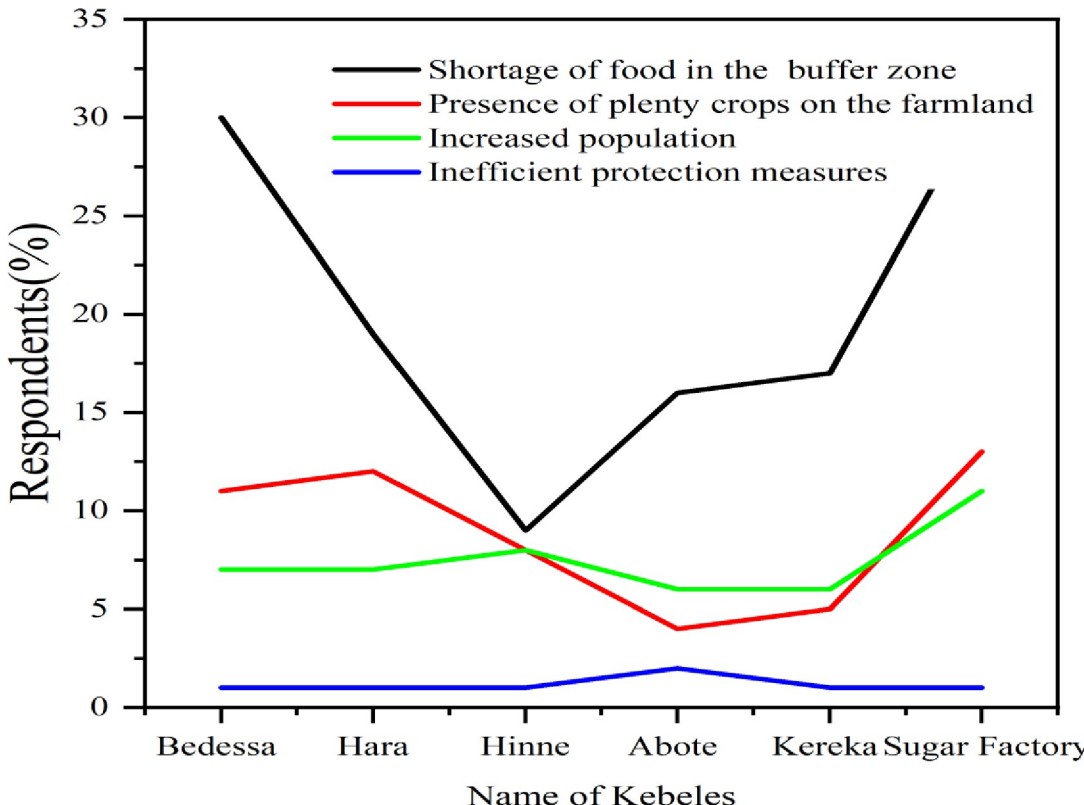

**Fig 3. People's perception on the driving factors of human hippopotamus conflict.**

**Table 4. Attitude of respondents towards hippopotamus conservation.**

| Variable | n | Negative (%) | Positive (%) | Neutral (%) | df | $X^2$ | p-value |
|---|---|---|---|---|---|---|---|
| Sex | | | | | | | |
| Male | 153 | 41.9 | 22.5 | 3.1 | | | |
| Female | 74 | 24.7 | 5.7 | 2.2 | 2 | 10.48 | 0.033[*] |
| Age (year) | | | | | | | |
| 18–24 | 33 | 5.7 | 8.8 | 0.4 | | | |
| 25–34 | 74 | 19.4 | 12.8 | 0.9 | 6 | 36.12 | 0.000*** |
| 35–50 | 90 | 30.8 | 5.7 | 2.6 | | | |
| >51 | 30 | 10.1 | 1.8 | 0.9 | | | |
| Education level | | | | | | | |
| Illiterate | 68 | 26 | 1.8 | 2.2 | | | |
| Primary school | 97 | 24.7 | 16.3 | 1.8 | 6 | 90.2 | 0.000*** |
| Secondary school | 10 | 1.3 | 2.2 | 0.9 | | | |
| College and above | 52 | 4.8 | 17.6 | 0.4 | | | |
| Villages distance from sanctuary | | | | | | | |
| Bedessa Didessa (1km) | 49 | 87.8 | 10.2 | 2 | | | |
| ADSF (1km) | 56 | 84 | 14.3 | 1.8 | 10 | 59. 9 | 0. 000*** |
| Abote Didessa (1.1-2km) | 28 | 85.7 | 7.14 | 7.14 | | | |
| Kereka Didessa (2.1-3km) | 30 | 70 | 16.7 | 13.3 | | | |
| Hara (3.1-4km) | 39 | 41 | 51 | 7.7 | | | |
| Hinne (4.1-5km) | 25 | 32 | 64 | 4 | | | |
| Livelihood source | | | | | | | |
| Mixed agriculture | 163 | 56.4 | 11.5 | 3.9 | | | |
| Only farming | 8 | 2.6 | 0 | 0.9 | 4 | 66.6 | 0.000*** |
| Wardens/employee | 56 | 3.5 | 5.3 | 0 | | | |
| Livestock size | | | | | | | |
| 1–10 | 170 | 52.9 | 16.7 | 5.3 | | | |
| 11–20 | 37 | 10.1 | 6.2 | 0 | 4 | 18.9 | 0.016* |
| 21–30 | 20 | 3.5 | 5.3 | 0 | | | |

*Significance at 95% confidence interval

*ADSF = Arjo Dhidhessa Sugar Factory

human injuries (13 were fatal) (see S3 Appendix). Until recently, some soldiers assigned to guard the state-owned Dhidhessa Sugar Factory shot and killed hippo as a strategy to deter them from entering the cane farm (S4 Fig in S2 Appendix). As this activity is illegal by law in the country, tracking and recovering the dead bodies of the animals was hard.

## Mitigation strategies experienced in the area

In this study area, farmers used multiple human-hippo conflict mitigation strategies. Guarding farms from elevated vantage points and shouting at approaching hippo was the most used (59.5%, n = 135) and considered the most effective crop protection strategy against hippo damage in the area. Fencing farms as physical barrier (15.9%, n = 36) and night time fire burning around the edge of farms (10.6%, n = 24) were also implemented strategies to prevent or lower crop damage by hippo, and the variation was not significant ($x^2 = 22.26$, df = 3, p>0.05) (Fig 4). In lesser extents, building deep trenches around the edges of farms and major hippo routs were also experienced (Fig 5). Some informants reported that they used to kill hippos as a strategy to prevent crop loss but realised that retaliation by injured hippo and the herd had

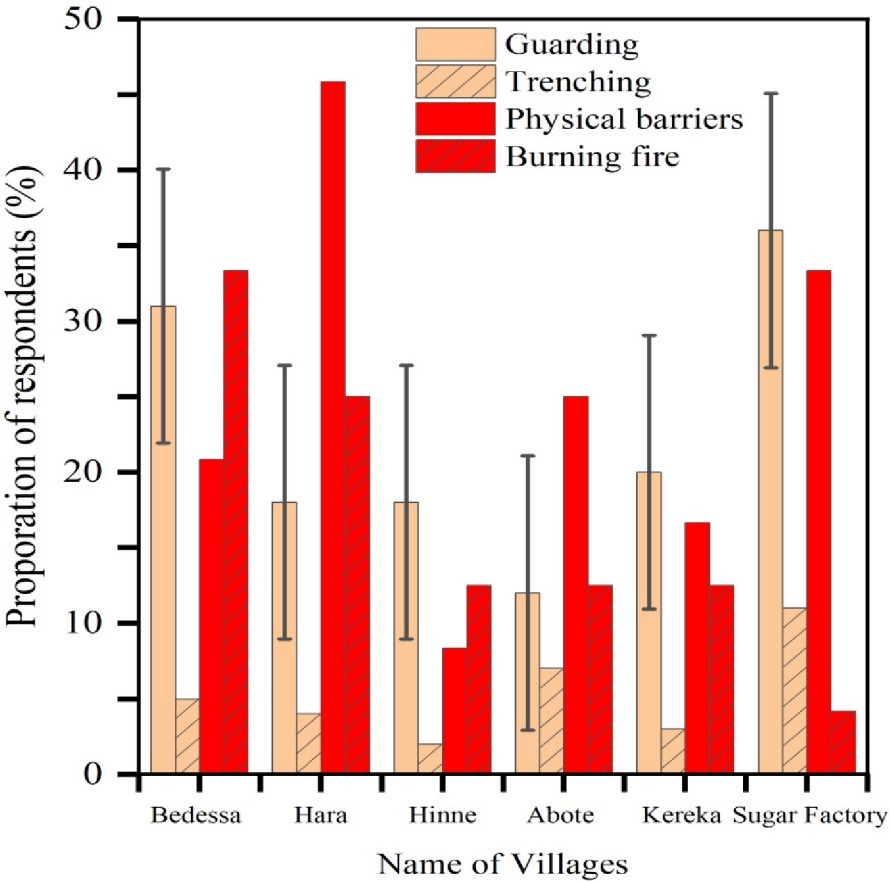

**Fig 4. Mitigation measures used by local population to prevent crop damage caused by hippopotamus.**

more severe consequences than the strategy was least implemented. No compensation scheme established or experienced for the lost and damaged crops, livestock loss, and human injuries or deaths as a result of the conflict. Only very few compensation experience were reported only for casualties in the Arjo-Dhidhessa sugarcane farm.

## Discussion

### Damage caused by hippopotamus

This study revealed that there is high human-hippo conflict in the Dhidhesa wildlife sanctuary. While crop loss and destruction represented the major human-hippo conflict, physical threats, human injuries and fatalities were also common. Land use changes within hippopotamus habitats may have increased the conflict incidences in addition to crop loss and physical threats. Human-hippo conflicts were found to have seasonal trend, with an annual peak occurring during the seedling from mid- and late- crop vegetative growth stages in the majority of the study sites. Similar findings were reported from farms in the Mara River area [19], Arjo-Dhidhesa Sugar factoryin Ethiopia [20], and Chebera Churchura National Park in Ethiopia [16].

Studies on the behavioural ecology of hippo revealed that they are habituated to leaving their pools in the early evening, after sun-set and back to pools in the early morning, little after sunrise [21]. Most of the animals out- of water life activities are during the night, where tracking every destructive activity relying on vegetation remains after grazing, hoof marks, trampled

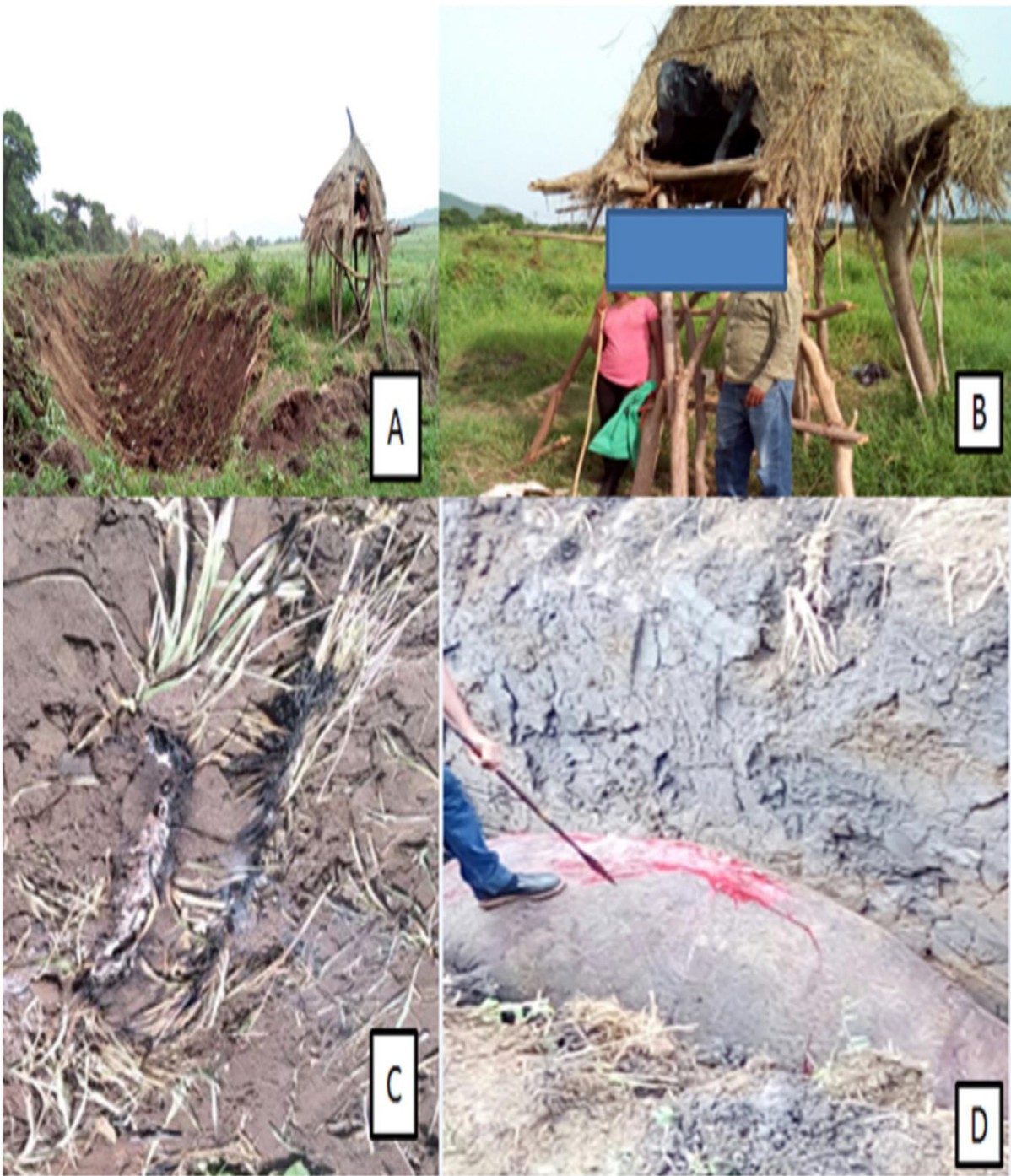

**Fig 5.** Different mitigation strategies using at the Dhidhessa wildlife sanctuary to prevent hippo attacks: (a) hippo trench, (b) guarding, (c) fire, and (d) killing hippo on the hippo trench.

vegetation, and dung piles left by the animal. Such behaviour makes the mitigation strategies less effective. Our data show that, farms situated closer to the river are most affected than those located at a distance. Similar observations were reported elsewhere [16, 22].

It is clear that the lack of a buffer zone between the sanctuary, river banks, and farmlands was likely the cause of the highest crop loss incidence among nearby households. Furthermore, the lack of a buffer zone may facilitate more interactions between hippos and farmlands in the present study area, hence raising the risk of crop destruction. Comparable research conducted in other parts of the world has shown that local communities that are near protected areas are more vulnerable to crop damage than those that are farther away [24–26].

Reports show that wildlife species involved in raiding crops show preference for specific crop types and different growth stages [23, 24]. Few reports are available to generalise crop preference tendencies of hippo, for instance, rice [21] and maize [25]. Evidences during this survey revealed no clearly established trend of crop preference for the hippo population; however, sugarcane and maize are the most damaged crops, most during the early stages of vegetative growth. Similar observation was reported by [26].

The vulnerability of a farm to hippopotamus attack varies based on a number of circumstances and factors [16, 22, 27]. Likewise, in the present study area, farm size, distance of the farm from hippo infested river, farms on or near the movement rout of the animals, crop growing season, proximity of the farm to other farms or settlements, and the available farm guarding techniques were mentioned as important factors. As compared to crop loss reports in Africa [22], the severity is very high in the Dhidhessa wildlife area. This demonstrated that there was a high incidence of crop raiding in the study area, which is dependent on a number of factors like the level of human activity on the farm, the accessibility of resources for hippo, and the cultivation of crops in relation to the sanctuary location.

In the present observation, the impacts of hippo on the farm are two fold greater than those caused by most known crop pests. Besides heavy grazing, hippo trampled fields, destroying more crops than fed, which is more impactful when done at the later growth stage of crops (where recovery is hardly possible) [28] reported a similar observation elsewhere.

Dhidhessa wildlife sanctuary is located within the Tsetse belt of the country [29], where pastoralism and large -scale cattle farming sometimes become challenges. For the ecology of Tsetse fly (a vector for the African trypanosomiasis (AT), resting and waiting for host in areas with high wildlife excreta (especially urine- sprayed ground) [30], and as hippo is the second highly favored mega herbivore [31], the reported impacts of hippo in transmitting the deadly infection to livestock of the area becomes a real challenge. Although reports on the impacts of the AT on hippo are rare, these may serve as a reservoir for the infection from where the livestock contracted the deadly infection. In the area, only native cattle breeds have a history of resisting AT infection.

## Trend and driving force of human hippo conflict

Human-hippo conflicts in hippo range countries are continually growing. In most instances, draining wetlands for dry season farming, fragmenting hippo habitats for small and large-scale agricultural practices, and restricting the free- ranging habits of hippo among habitats patches (for settlement and construction) are among the reasons for the current trends [22, 32, 33]. Before four decades, the Dhidhesa Wildlife Sanctuary was an ideal home to most wildlife species and specifically the best habitat for hippo. Initially, the government used the extensive lowland grassland area for extensive settlement for draught -affected societies from the North and east part of Ethiopia. This activity shifted the wildlife area from the south and the western part. Then, the state- owned Argo-Dhidhessa Sugar Factory and mechanized sugarcane plantation took several thousand hectares of land from the northern parts of the sanctuary. In the northern and northeastern part of the river, practically no space is left for wildlife or hippo. This situation created a high opportunity for repeated human-hippo contacts, triggering

unusual aggressive responses that the later ended in severe injuries and human fatalities. Similar experiences were reported by [34].

## Consequences of conflict on both human communities and hippo populations

[35] Phrased "human-animal conflict" as circumstances where the interest of human and wildlife enter into conflict that has a negative impact on either humans or wildlife. Accordingly, wildlife poses a threat to humans' livelihoods as a result of property loss and the risks on personal safety [36]. On the other side, human activities caused habitat loss and degradation to the extent that they resulted in wildlife population decline, migration, altering wildlife behavior, range reduction, and local extirpation [28, 37, 38]. The societies in the Dhidhessa wildlife sanctuary and the hippo population of the area well experienced the stated impacts of the conflicts that arose and intensified in in the latest decades. Extensive settlements and uncontrolled encroachments by smallholder and government- initiated large- scale commercial farming severely impacted wildlife habitat, the connectivity of habitats, and ranging routs of the animals, on the one hand, and crop and other property losses, human injuries, and fatalities, on the other hand. Adverse reactions from both sides (household-level food insecurity, complete crop loss, human fatalities, including head of the family) and the massive massacre of hippo become the highest concerns of this study.

## People's attitude towards hippo conservation

The way people feel about wildlife has huge impact on the successfully protection and conservation of a specific species or wild animals in general [39, 40], leading to the lethal eradication of potentially dangerous species [41]. Results from the present study area clearly show similar situations in that the majority of the respondents, for life safety reason, gradually developed strong fear for the presence and encounters of hippo and, hence, were un-happy for their conservation. Although at this stage, retaliation was not the preferred alternative to avoid hippo, the current trend is frustrating. On the other hand, socio-demographic factors, such as gender [40, 42–44], age [45], and level of education are crucial in contributing to the tolerance, acceptance, and coexistence of humans and the wildlife, including hippo [45, 46]. Disparities between gender, age, and educational level towards the conservation of hippo were also observed. In this regard, more males, younger generations and relatively educated individuals show interest in the animal and its conservation. This is an encouraging observation because the key reason for this finding in the study area is that individuals with higher levels of formal education might be more aware of issues related to conservation because they interact with educational institutions and are exposed to more media. On the other hand, young women in the study area were driven by a variety of factors to improve access to education, including independence, the ability to make their own decisions, empowerment, employment, and the benefits of natural resources, in order to support wildlife conservation programs. This outcome was somewhat encouraging, even if the study's participants admitted that the benefits were minor (for example, DWS does not offer any financial advantages related to hippos through ecotourism, education, or ecological benefits of hippos) [47–49].

## Mitigation measures of human hippo conflict

Effectively designed and implemented human wildlife conflict mitigation and management strategies significantly lower the impacts of human wildlife conflicts, and leading towards co-existence [33, 50, 51]. To prevent hippo attacks on crops in the study area, local people adopt a number of mitigating techniques. Farm guarding, fencing (physical obstacles), digging hippo

trenches, burning fire, and scaring hippos by throwing stones were frequently used mitigation techniques. Although most of these strategies are old, laborious and need require continuous human involvement, they are considered effective and experienced by most traditional societies elsewhere [51, 52]. There is no crop or livestock loss compensation scheme in Ethiopia in general and in the current study area in particular. Otherwise, this is the most effective measure to contribute to better tolerance in the community and mitigation of HWC [44, 53]. During this study, only the Arjo Dhidhessa sugar factory established a scheme to compensate for injuries or deaths of wardens or other employees as a result of hippo attacks.

The worst control strategy observed in Arjo-Dhidhessa Sugarcane farm during this study was that workers produced hippo trenches (a dip- narrow furrow dug around the farm entrances) that are used as hippo traps. The deep dug furrow is covered by grass, so that appears to be normal ground for hippo herd, where the leader is fallen into and unable to move anywhere thereafter. The locals then use traditional weaponry to kill the trapped animal. The hippopotamus is killed using this method as a trap (Fig 6). Similar experiences were reported by [21]. The trenches excavated along the Dhidhessa River shore were specifically designed to trap hippos, since once inside, they are unable to escape and are hence vulnerable to attack. For the hippo trench to be functional, it must be constructed in a way that prevents hippo access to sugarcane farms while still allowing them to enter and exit waterways. Hippos should also have access to open grassland habitats in the trench so they can forage there before returning to the river. According to [33, 54], all of these techniques are too archaic compared to the usage of barbed wire fences, electric wire fences, innocuous explosive materials, and other repellants used to stop wild animal attacks on farms. It is therefore advisable to employ a number of techniques to lessen the likelihood that wildlife may grow tolerant of any one particular one.

## Conclusions and implications for hippo conservation

The findings of the current study make it abundantly evident that there was significant conflict between the hippo and human populations in the studied area. The findings revealed that crop raiding is the major cause of conflict between humans and hippopotamuses in the study area. Crop damage, physical threats, human injury, livestock mortality, and hippos' mortality are the main consequences of high conflict (Fig 2 and S1 Fig in S2 Appendix). Lack of food in the forest or buffer zone, an abundance of crops on farmland, human population growth, negative attitudes of local people towards hippo, inefficient hippo protection measures and activities such as deforestation, inappropriate site selection for investment in protected areas, habitat degradation by expansion of agriculture and settlement, competition with livestock, and killing and chasing of hippopotamuses activities together have led to increased human encroachment on previously wild and uninhabited areas (Figs 3, 6 and S2 Fig in S2 Appendix). In addition, human action highly agitates hippos and frequently results in their deaths. If this trend is not reined in and the conflicts are not effectively managed, the number of hippopotamus in the study region may soon become unsustainable (S4 Fig in S2 Appendix). Locals in the research area did not like the conservation of hippos. In terms of hippo conservation, this is really detrimental. A well-designed mitigation strategy in the study area is therefore essential to improving the coexistence of people and hippos there. Furthermore, the conservation strategy should also entail community participation in the coordination of local and district management offices, as well as community education, factory workers, sugarcane guardians, and awareness campaigns [55]. The primary subjects of these educational and communication programmes should be the cultural importance of hippos to communities, their role as ecosystem engineers [56], and their worth as a tourist attraction. We also suggest conducting surveys of the

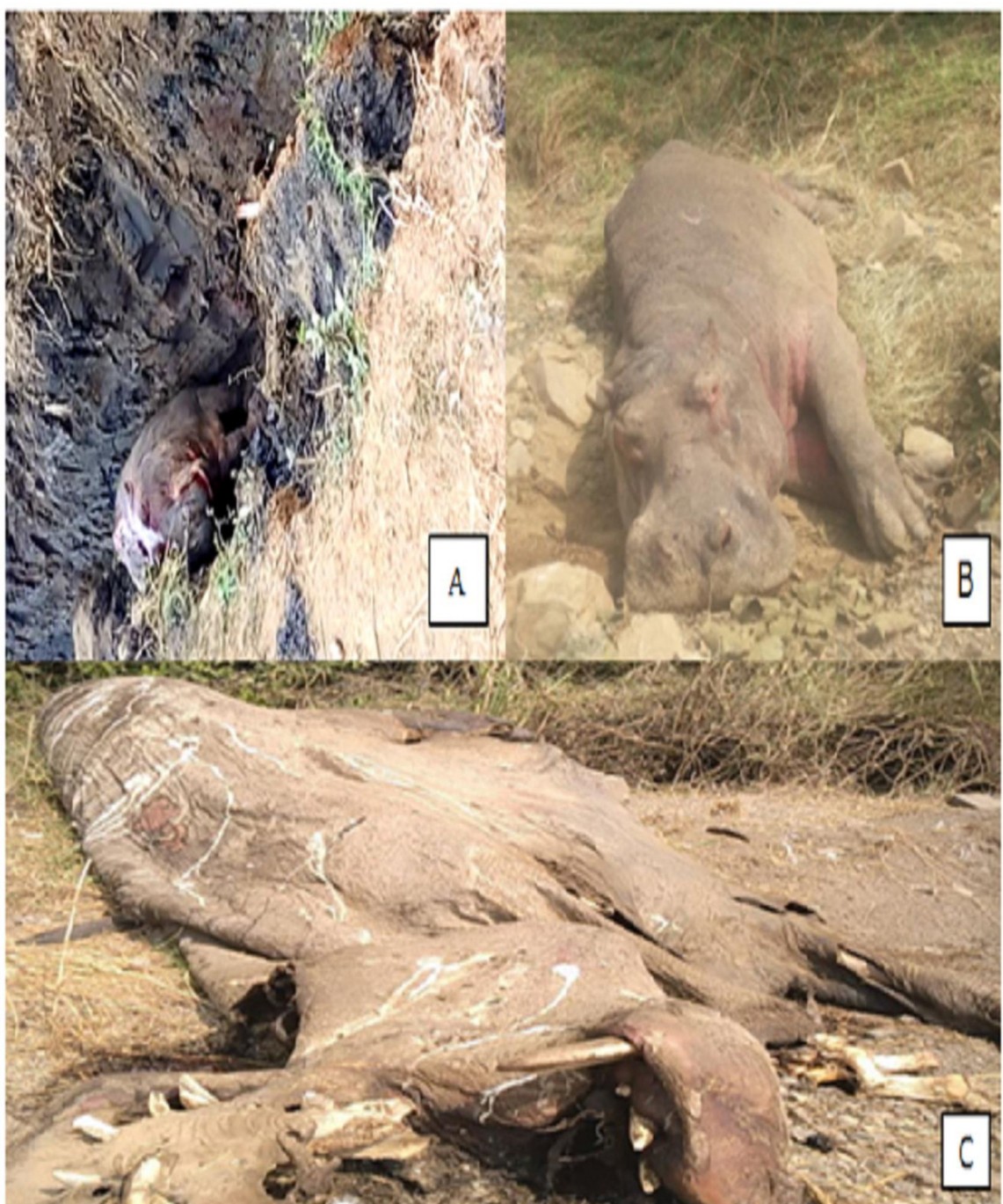

**Fig 6.** Dead hippos on the farm around Dhidhessa River: (a) killing hippo on the trench, (b and c) killing hippo on the farm.

population, ecology, and distribution of hippos, examining the role of hippos in disease transmission to cattle, conducting experimental tests on how hippos respond to deterrent methods in order to implement the most effective mitigation approaches, and quantifying the nature of anthropogenic-related threats to hippos and their habitats in the area.

## Supporting information

**S1 Appendix. Household questionnaire guide for human-hippopotamus conflict.**
(DOCX)

**S2 Appendix. Supplementary.**
(DOCX)

**S3 Appendix. Descriptive statistics of variables.**
(XLSX)

## Acknowledgments

We thank to IDEA-WILD for their material support. We appreciate the management of the Arjo Dhidhessa Sugar Factory. All the supports from the local communities were also sincerely acknowledged.

## Author Contributions

**Conceptualization:** Girma Gizachew Tefera, Tadesse Habtamu Tessema, Tibebu Alemu Bekere, Tariku Mekonnen Gutema.

**Data curation:** Girma Gizachew Tefera.

**Formal analysis:** Girma Gizachew Tefera.

**Investigation:** Girma Gizachew Tefera, Tariku Mekonnen Gutema.

**Methodology:** Girma Gizachew Tefera, Tadesse Habtamu Tessema, Tibebu Alemu Bekere, Tariku Mekonnen Gutema.

**Resources:** Tadesse Habtamu Tessema.

**Supervision:** Tadesse Habtamu Tessema, Tariku Mekonnen Gutema.

**Visualization:** Tadesse Habtamu Tessema, Tibebu Alemu Bekere.

**Writing – original draft:** Girma Gizachew Tefera.

**Writing – review & editing:** Girma Gizachew Tefera, Tadesse Habtamu Tessema, Tibebu Alemu Bekere, Tariku Mekonnen Gutema.

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
