## [Decision Letter · Decision Letter 0]

19 Feb 2024

PONE-D-23-34333Human-common hippopotamus (Hippopotamus amphibius)-conflict in the Dhidhessa Wildlife Sanctuary and its surrounding, Southwestern EthiopiaPLOS ONE

Dear Dr. Tefera,

Thank you for submitting your manuscript to PLOS ONE. Please accept my apology for the length of time it took to be able to reach a decision on this manuscript.  After careful consideration, we feel that it has merit but does not fully meet PLOS ONE’s publication criteria as it currently stands. Therefore, we invite you to submit a revised version of the manuscript that addresses the points raised during the review process.

The authors present findings from a human-hippo conflict study that has the potential to provide a valuable contribution to the literature. The authors provide results from descriptive statistical analyses, using programs for analysis that are not broadly known making it more important for the authors to cite the sources of these programs. Reviewer #2 provided suggestions for introducing more analytical rigor. There are straightforward multivariate approaches that can allow the authors to more clearly understand the factors that influence attitudes. However, the addition of multivariate approaches should not come at the expense of accuracy; the authors would need to their comfort level with the assumptions and limitations of any new analysis conducted, especially with the use of point-and-click statistics programs. Reviewer #2 provides additional feedback that will be helpful when revising your manuscript. Please note that submission of a revised manuscript addressing reviewer comments does not guarantee acceptance. The journal office looked into the claims from Reviewer #1 and did not find evidence of plagiarism; however, if there are similar publications, please cite appropriately in the manuscript. Please ensure that your decision is justified on PLOS ONE’s publication criteria and not, for example, on novelty or perceived impact.

We look forward to receiving your revised manuscript.

Kind regards,

Stephanie S. Romanach, Ph.D.

Academic Editor

PLOS ONE

Journal Requirements:

4. In the online submission form, you indicated that the data used in this study are available by the authors upon reasonable request

5. We note that Figures 1 and 2 in your submission contain [map/satellite] images which may be copyrighted. All PLOS content is published under the Creative Commons Attribution License (CC BY 4.0), which means that the manuscript, images, and Supporting Information files will be freely available online, and any third party is permitted to access, download, copy, distribute, and use these materials in any way, even commercially, with proper attribution. For these reasons, we cannot publish previously copyrighted maps or satellite images created using proprietary data, such as Google software (Google Maps, Street View, and Earth). For more information, see our copyright guidelines: http://journals.plos.org/plosone/s/licenses-and-copyright.

a. You may seek permission from the original copyright holder of Figures 1 and 2 to publish the content specifically under the CC BY 4.0 license.  

Reviewers' comments:

Reviewer's Responses to Questions

**Comments to the Author**

1. Is the manuscript technically sound, and do the data support the conclusions?

Reviewer #1: No

Reviewer #2: Partly

2. Has the statistical analysis been performed appropriately and rigorously? 

Reviewer #1: No

Reviewer #2: No

3. Have the authors made all data underlying the findings in their manuscript fully available?

Reviewer #1: No

Reviewer #2: Yes

4. Is the manuscript presented in an intelligible fashion and written in standard English?

Reviewer #1: No

Reviewer #2: Yes

5. Review Comments to the Author

Reviewer #1: The manuscrbite is not writenn well. The data collection method and analysis are not good and scientific. The findings are not presented properly. Majority of the statments are copied from someones paper.

Reviewer #2: Main Comments:

Here, the authors investigate the extent of human-hippo conflict and the perception that local inhabitants have about the conflict between hippos and people in the Dhidhesa Wildlife Sanctuary, and what this conflict means for hippo conservation in the area. The authors make use of questionnaire surveys, focus group discussions and key informant interviews as their primary data source for the study. These results have the potential to provide key baseline information on human-hippo conflict, which has generally been understudied. Overall, I enjoyed reading the paper which could make an important contribution to the field. However, I did note several concerns during the paper review process.

While the paper covers an important and interesting topic, the writing style, including grammar, sentence construction and sentence linking, requires improvement. It was a challenge at times to assess the manuscripts content because of this. I suggest a careful read through and gleaning of the article, and recommend obtaining the assistance of a scientific editorial service to improve the manuscripts readability. I have tried where possible to provide some grammatical feedback, but I have not been able to include all my notes and observations.

Specific comments:

Most of the data is presented in the form of descriptive statistics, which is to be expected for a questionnaire-type study. However, I do think the authors could introduce a more rigorous analytical routine to provide adequate support for the conclusions made in the discussion.

Some of the figures are incorrectly referenced in the main text. This needs to be inspected and corrected throughout the manuscript. What is the reasoning for including Figures versus Plates? The authors should also refrain from stretching the images. Images could be grouped into a single figure labelled a-e (or however many figures there are) that occur alongside each other (double column) on a single page.

Line 1: Consider replacing with “Human-hippo (Hippopotamus amphibius) conflict...”

Line 30: What about rhinos? I would rephrase to say "…one of the most endangered megaherbivores in Africa…"

Line 67: “…listed as vulnerable…” instead of “tagged vulnerable”

Line 88-92: This paragraph requires rewording; it is a challenge to decipher what the authors are trying to say.

Line 100-117: The “Ethical statement” should not be the first paragraph of the Methods Section. This should either come later in the methods or at the very end of the manuscript, towards the acknowledgements section. There also appears to be another “Methods” section below “Materials and Methods” – try be more specific e.g., Study Design or Survey Approach

Line 133-135: Not clear whether the reserve was designated for its high diversity or large hippo populations? This needs to be made clearer.

Line 142: “…human-dominated ecosystem…” rather than “…human ecosystem…”

Line 157: “Focus group discussions” do not need to be abbreviated if it is only mentioned a handful of times in the manuscript, but if it is abbreviated, then be sure to abbreviate as FDG when this is first used, and then use the abbreviation somewhere in the main text.

Line 159: “…total HUMAN population…”? Unclear what population the authors are referring to.

Line 162-164: remove, “…for the randomization…” as the randomization idea is mentioned again in the same sentence

Line 165: It would be helpful to briefly describe what a semi-structured questionnaire is?

Line 168: “…utilized in the area…” rather than “…experienced…”

Line 170: What is the Kobo Toolbox? This needs to be explained after is introduced to the reader. Include a sentence of what this is and why it was used.

Line 183: Describe which program was used to conduct the analysis only at the end of the paragraph.

Line 184-186: Alternatively, “…To determine the effect of distance, village and perception on HHC…” etc.

Line 192: 43% is not “most”. Alternatively, “Less than half of the community have a primary level of education (43%).”?

Line 252-273: While this paragraph is insightful, it needs to be reworded to be less speculative. The authors could state that “…of the people interviewed, there was a strong trend of increased conflict from 2014, which so happens to coincide with encroachment onto their habitat…”. Or whatever makes most sense from the data available.

Line 303: Refrain from using the word “proved” and instead use “…we show…” or “…our data show…”

Line 304-307: An insightful finding from the study, but again it needs to be re-written and re-worded to make it understandable.

Line 319-320: Do you mean that the population size of hippos in the DWS is higher than reported elsewhere in Africa?

Line 336: An important reference to be included here and elsewhere in the manuscript: Van Houdt, S. & Traill, L.W. (2022). A synthesis of human conflict with an African megaherbivore; the common hippopotamus. Frontiers in Conservation Science 3, 954722.

Line 372-376: Which is a strong motivating factor to improve access to education for young woman in the community?

Line 424: Reference: Voysey et al (2023). Are hippos Africa’s most influential megaherbivore? A review of ecosystem engineering by the semi-aquatic common hippopotamus. Biological Reviews

Table 1: It is unclear what “With Farms” and “Without Farms” means here? Describe what this means in the table heading.

Figure 4: Does this figure describe to what extent the respondents are in support of hippo conservation? It is not perfectly clear that this is indeed the case.

6. PLOS authors have the option to publish the peer review history of their article (what does this mean?). If published, this will include your full peer review and any attached files.

Reviewer #1: No

Reviewer #2: No

---

## [Author Response · Author response to Decision Letter 0]

7 Apr 2024

Subject: Responses to the reviewers’ comments

Dear Dr. Stephanie S. Romanach, 

We are very grateful for your and reviewers’ critical and constructive comments regarding our manuscript numbered PONE-D-23-34333, entitled “Human-common hippopotamus (Hippopotamus amphibius)-conflict in the Dhidhessa Wildlife Sanctuary and its surrounding, Southwestern Ethiopia”. Please find re-submitted (the revised version) of our manuscript in which we have carefully considered and addressed all comments raised by the referees. The revised version of the manuscript includes many improvements and summarized below. Moreover, the revised manuscript is presented using track-change with red color. We hope our responses are satisfactory and you will find this version suitable for publication in Journal of PLOS ONE. We look forward to hearing from you.

Dear Academic Editor, please note the following points

1. Our responses to the reviewers’ comments are provided 

2. The line numbers with red colors (line#) refer to the improved version of the revised manuscript 

Best regards, 

Corresponding author 

Response to Academic editor #1

1. Comment 1(General comment), the reviewers suggested that the manuscript be corrected for problems of language usage, spelling, grammar, data collection methods and analysis and I found many such problems that were not addressed in the revision. Some of the revisions introduced new problems as well. To help improve the quality of the manuscript, I include a list of specific revisions that are needed below. 

Response: Thank you very much for your suggestion. We agree with the comment and the whole section is improved by reducing or amending some of the words, phrases or sometimes sentences.

• The manuscript should meet PLOS One's style requirements.

• Manuscript for language usage, spelling, and grammar.

• Grant information in the ‘Funding Information’ and ‘Financial Disclosure’ sections does not match.

• Data availability by the authors upon reasonable request.

• Figures 1 and 2 in your submission contain [map/satellite] images, which may be copyrighted. We made improvements to the issues mentioned above.

2. Comment 2, Line 1: Consider replacing with “Human-hippo (Hippopotamus amphibius) conflict..."

 Line 30: What about rhinos? I would rephrase to say "…one of the most endangered mega herbivores in Africa…" 

 Line 67: “…listed as vulnerable…” instead of “tagged vulnerable” 

 Response: Thank you very much for your suggestions. We agree with the comment and made improvements on Line 1: replacing with “Human-hippo (Hippopotamus amphibius) conflict” 

Line 30: "currently tagged" changed to” one of the most endangered mega herbivores in Africa" and Line 67: “tagged vulnerable” changed to “listed as vulnerable” Using a track change from line# 1, 29 &67, respectively

3. Comment 3, Line 88-92: “Unlike most areas in Ethiopia, the culture and actors of HWC in the Dhidhesa wildlife sanctuary is different. Like most hippos dominated human encroached ecosystem, there are plenty of reports for the existence and severity of human-hippo conflict in the area. However, there is no information regarding the causes, extent, and consequences of HHC on both actors in the area are not known. “This paragraph requires rewording; it is a challenge to decipher what the authors are trying to say“ 

Response: Thank you very much for your suggestions. We agree with the comment and made correction using a track change from Line# 89-92; “Recently, Dhidhessa Wildlife Sanctuary has not provided the intended objectives for which it was established. Subsequently, large mammals that occurred in the area migrated into the nearby forest patches. But the hippo is one of many large mammals that have been living in the Dhidhessa River valley. However, in the study area, no comprehensive research has been done on the causes, extent, and consequences of HHC in the southwestern regions of Ethiopia“.

4. Comment 4, Line 100-117: The “Ethical statement” should not be the first paragraph of the Methods Section. This should either come later in the methods or at the very end of the manuscript, towards the acknowledgements section. There also appears to be another “Methods” section below “Materials and Methods” – try be more specific e.g., Study Design or Survey Approach

Response: Thank you very much for your suggestions. But the guidelines of PLOS ONE stated that “ethical statement” should be the first paragraph under methods Section. However, we agree with the comment and made correction “Materials and Methods” We specified to “Site selection and sampling design” from Line# 142 

5. Comment 5, Line 133-135: Not clear whether the reserve was designated for its high diversity or large hippo populations? This needs to be made clearer. 

Response: Thank you very much for your comments/suggestions. We agree with the comment and made correction by rewriting the paragraph using a track change from Line# 133-141:” Even so, the area is not officially protected at the moment because it is heavily utilized by the private sector and local communities for mechanized monoculture cane farms, subsistence farming, husbandry, and other economic activities that put them in direct conflict with the hippo and other wildlife species in the area. Nevertheless, it remains an important area for biodiversity conservation in the southwestern parts of Ethiopia. In addition to hippos, DWS and the adjacent surroundings are home to a diversity of animal species, including primates (Papio anubis and Cercopithecus mitis), Buffalo (Syncerus caffer), Lion (Panthera Leo) Guerza (Colobus guerza),Warthog (Phacochoerus africanus), Bushpig (Potamochaerus larvatus) and Spotted Hyena (Crocuta crocuta) [ 20]”

6. Comment 6, Line 142: “…human-dominated ecosystem…” rather than “…human ecosystem…”

Response: Thank you very much for your suggestions. We agree with the comment and made correction rewrite using a track change from Line# 133-141 

7. Comment 7, line 157: “Focus group discussions” do not need to be abbreviated if it is only mentioned a handful of times in the manuscript, but if it is abbreviated, then be sure to abbreviate as FDG when this is first used, and then use the abbreviation somewhere in the main text.

Line 159: “…total HUMAN populations…”? Unclear what population the authors are referring to.

Line 162-164: remove, “…for the randomization…” as the randomization idea is mentioned again in the same sentence

Line 165: It would be helpful to briefly describe what a semi-structured questionnaire is?

Line 168: “…utilized in the area…” rather than “…experienced…”

 Response: Thank you very much for your comments. We agree with the comment and made correction from Line# 157,159,162-164,165 and 168 “As you indicated, the data collection methods and analysis paragraph are organised poorly." So, based on your comments and suggestions using a track change from Line# 163-207, we re-write or revised the whole of those portions.

8. Comment 8, Line 170: What is the Kobo Toolbox? This needs to be explained after is introduced to the reader. Include a sentence of what this is and why it was used.

 Response: Thank you very much for your comments and suggestions. We agree with the comment and made correction using a track change from Line# 172-174. As you indicated, KoBoToolbox is a suite of tools for field data collection for use in remote areas or where the internet facility is uncertain. Moreover, it is a very easy tool to use and copy the stored data (https://support.kobotoolbox.org/). 

9. Comment 9, Line 183: Describe which program was used to conduct the analysis only at the end of the paragraph. 

 Response: Thank you very much for your comments. We agree with the comment and made correction using a track change from Line# 216-217 (we added at the end of the paragraph) 

10. Comment 10, Line 184-186: Alternatively, “…To determine the effect of distance, village and perception on HHC…” etc.

 Response: Thank you very much for your comments and suggestions. We agree with the comment and made correction using a track change from Line# 209-217 (as indicated that, we rewriting the analysis section) 

11. Comment 11, Line 192: 43% is not “most”. Alternatively, “Less than half of the communities have a primary level of education (43%).”?

Response: Thank you very much for your suggestions. We agree with the comment and made correction. We rewrite the demography and Socio-economic activities from Line# 220-229 with track changes. 

12. Comment 12, Line 252-273: While this paragraph is insightful, it needs to be reworded to be less speculative. The authors could state that “…of the people interviewed, there was a strong trend of increased conflict from 2014, which so happens to coincide with encroachment onto their habitat…”. Or whatever makes most sense from the data available.

Response: Thank you very much for your comments and suggestions. We agree with the comment and made correction using a track change from Line# 296-320 (this section was reworded by using the data of the interviewee, as suggested) 

13. Comment 13, Line 303: “Refrain from using the word “proved” and instead use “…we show…”or“…our data show…”

Response: Thank you very much for your comments. We agree with the comment and made correction using a track change from Line# 350 

14. Comment 14, Line 304-307: An insightful finding from the study, but again it needs to be re-written and re-worded to make it understandable.

Response: Thank you very much for your comment. We agree with the comment and made correction using a track change from Line# 352-357 ("involving" re-write and re-worded as suggested). 

15. Comment 15, Line 319-320: Do you mean that the population size of hippos in the DWS is higher than reported elsewhere in Africa?

Response: Thank you very much for your comment. We agree with the comment and made correction using a track change from Line# 369-372 (replaced by: “This demonstrated that there was a high incidence of crop raiding in the study area, which is dependent on a number of factors like the level of human activity on the farm, the accessibility of resources for hippo, and the cultivation of crops in relation to the sanctuary location”). 

16. Comment 16, Line 336: An important reference to be included here and elsewhere in the manuscript: Van Houdt, S. & Traill, L.W. (2022). A synthesis of human conflict with an African mega herbivore; the common hippopotamus. Frontiers in Conservation Science 3, 954722. 

Response: Thank you very much for your comments and suggestions. We agree with the comment and made correction using a track change from Line# 390, 442 and 463

17. Comment 17, Line 372-376: Which is a strong motivating factor to improve access to education for young woman in the community?

Response: Thank you very much for your suggestions. We agree with the comment and made correction using a track change from Line# 429-435 ("The results of the study indicate that young women in the study area were driven by a variety of factors to improve access to education, including independence, the ability to make their own decisions, empowerment, employment, and the benefits of natural resources, in order to support their current and future families"). 

18. Comment 18, Line 424: Reference: Voysey et al (2023). Are hippos Africa’s most influential mega herbivore? A review of ecosystem engineering by the semi-aquatic common hippopotamus. Biological Reviews

Response: Thank you very much for your suggestions. We agree with the comment and made correction using a track change from Line# 487 

19. Comment 19, Table 1: It is unclear what “With Farms” and “Without Farms” means here? Describe what this means in the table heading.

Response: Thank you very much for your comment. We agree with the comment and made correction. We describe those values “with farms “mean having cultivated crops, while values “without farms “mean non-cultivated crops.

20. Comment 20, Figure 4: Does this figure describe to what extent the respondents are in support of hippo conservation? It is not perfectly clear that this is indeed the case.

Response: Thank you very much for your suggestions. We agree with the comment and made correction using a track change from Line# 265-271,272-287 ("People’s perceptions and attitudes towards hippo conservation under the analysis section were rewritten, and figure 4 was removed and replaced by table 4”).

---

## [Editor Report · Decision Letter 1]

16 Apr 2024

PONE-D-23-34333R1Human- common hippo (Hippopotamus amphibius)-conflict in the Dhidhessa Wildlife Sanctuary and its surrounding, Southwestern EthiopiaPLOS ONE

Dear Dr. Tefera,

Thank you for submitting your manuscript to PLOS ONE. After careful consideration, we feel that it has merit but does not fully meet PLOS ONE’s publication criteria as it currently stands. Therefore, we invite you to submit a revised version of the manuscript that addresses the points raised during the review process. 

Some of the review comments were not addressed in your response. For example, I see no response to the reviewer suggestion for a more rigorous and analytical approach nor about the figure stretching. I can see where you did and did not make edits but in the future you should respond to all reviewer comments. 

Some additional edits:

L 185 and 186 Kebele should be capitalized 

L 184 Woreda should also be capitalized 

L 209 change to “analyze”

L 215 What is text analysis?, Please describe 

L 236 do you mean trampled?

L 245 what percent loss is this? Or out of how many total cattle in the area?

L 257 remove “of responses”

L 260 remove “from time to time”, suggested revision to “over time”.

L 275 and other planes (e.g., L 346) “Hippo” should not be capitalized. Please change throughout 

L 281 change “better” to “more”

L 298 do you mean “three decades ago”?

L 351 river should be lower case, same on L 367

L 361 crop names should be lower case

L 426 age 5? The 5 may need to be removed or edited 

We look forward to receiving your revised manuscript.

Kind regards,

Stephanie S. Romanach, Ph.D.

Academic Editor

PLOS ONE
---

## [Author Response · Author response to Decision Letter 1]

24 Apr 2024

To: Stephanie S. Romanach, Ph.D.

 Academic Editor

 Journal of PLOS ONE 

Subject: Responses to the reviewers’ comments

Dear Dr. Stephanie S. Romanach, 

We are very grateful for your and reviewers’ critical and constructive comments regarding our manuscript numbered PONE-D-23-34333, entitled “Human-common hippo (Hippopotamus amphibius)-conflict in the Dhidhessa Wildlife Sanctuary and its surrounding, Southwestern Ethiopia”. Please find re-submitted (the revised version) of our manuscript in which we have carefully considered and addressed all comments raised by the referees. The revised manuscript is presented using track-change with red color and we attached also the manuscript without track. We hope our responses are satisfactory and you will find this version suitable for publication in Journal of PLOS ONE. We look forward to hearing from you.

Dear Academic Editor, please note the following points

1. Our responses to the reviewers’ comments are provided 

2. The line numbers with red colors (line#) refer to the improved version of the revised manuscript 

Best regards, 

Corresponding author 

Response to Academic editor #1

1. Comment 1(General comment), some of the review comments were not addressed in your response. For example, has the statistical analysis been performed appropriately and rigorously? 

Response: Thank you very much for your comments. We agree with the comment and made correction., Based on the comments and suggestions using a track change from Line# 143-217, 474,480,483,488 and 490 we have re-written or revised the whole of those portions.

• We have rewritten the demography and Socio-economic activities from Line# 220-229 with track changes

2. Comment 2, Figure 4: Does this figure describe to what extent the respondents are in support of hippo conservation? It is not perfectly clear that this is indeed the case.

Response: Thank you very much for your suggestions. We have made correction using a track change from Line# 265-271,272-287 ("People’s perceptions and attitudes towards hippo conservation under the analysis section were rewritten, and figure 4 was removed and replaced by table 4”). 

3. Comment 3, Line 185 and 186: Kebele should be capitalized 

 Response: Thank you very much for your suggestions. It has been changed as commented, on Line 185 and 186.

4. Comment 4, Line 184: Woreda should also be capitalized 

Response: Thank you very much for your suggestions. We have changed as commented, from Line# 184-186.

5. Comment 5, Line 209: change to “analyze” 

Response: Thank you very much for your suggestions. We have changed as commented, on Line 209.

6. Comment 6, Line 215: What is text analysis? Please describe 

Response: Thank you very much for your comments/suggestions. We agree with the comment and made correction by rewriting the words using a track change from Line# 214-215:” Information gathered from focus group discussions was analyzed to create a summary before being presented narratively”

7. Comment 7, Line 236: do you mean trampled? 

Response: Thank you very much for your suggestions. We agree with the comment and made correction to ‘trampled’ as commented on line 236.

8. Comment 8, line 245: what percent loss is this? Or out of how many total cattle in the area? Response: Thank you very much for your comments. We agree with the comment and made correction from Line# 245-246 “The results indicated that respondents throughout each surveyed village reared about 2368 cattle. Nearly 27.7% (n = 655) of the cattle from all the villages surveyed died as a result of the fatal infectious disease between 2013 and 2022’’.

9. Comment 9, Line 257: remove “of responses”

 Response: Thank you very much for your comments and suggestions. We have removed as commented on Line# 258. 

10. Comment 10, Line 260: remove “from time to time”, suggested revision to “over time”. Response: Thank you very much for your comments. We have changed as commented on Line# 261.

11. Comment 11, Line 275: L 275 and other planes (e.g., L 346) “Hippo” should not be capitalized. Please change throughout.

 Response: Thank you very much for your comments and suggestions. We have made correction as commented on Line# 275-276 and 347. 

12. Comment 12, Line 281: change “better” to “more” 

Response: Thank you very much for your suggestions. We have made correction on line 282. 

13. Comment 13, Line 298: do you mean “three decades ago”? 

Response: Thank you very much for your comments and suggestions. We agree with the comment and made correction using a track change on Line# 299 (this section was reworded to two decades represent a period of 20 years).

14. Comment 14, Line 351: river should be lower case, same on L 367 

Response: Thank you very much for your comments. We have changed as commented on Line# 352,368. 

15. Comment 15, Line 361: crop names should be lower case. 

Response: Thank you very much for your comment. We agree with the comment and have made correction on Line# 362.

16. Comment 16, Line 426: age 5? The 5 may need to be removed or edited. 

Response: Thank you very much for your comment. We agree with the comment and made correction on Line# 427 (edited by: “citation” [45]).

---

## [Editor Report · Decision Letter 2]

30 Apr 2024

Human- common hippo (Hippopotamus amphibius)-conflict in the Dhidhessa Wildlife Sanctuary and its surrounding, Southwestern Ethiopia

PONE-D-23-34333R2

Dear Dr. Tefera,

We’re pleased to inform you that your manuscript has been judged scientifically suitable for publication and will be formally accepted for publication once it meets all outstanding technical requirements.

Kind regards,

Stephanie S. Romanach, Ph.D.

Academic Editor

PLOS ONE
---

## [Editor Report · Acceptance letter]

4 May 2024

PONE-D-23-34333R2 

PLOS ONE

Dear Dr. Tefera, 

I'm pleased to inform you that your manuscript has been deemed suitable for publication in PLOS ONE. Congratulations! Your manuscript is now being handed over to our production team.

Kind regards, 

on behalf of

Dr. Stephanie S. Romanach 

Academic Editor

PLOS ONE